# Continual Graph Learning for Thermal Analysis of Composite Materials under Interface Variations

## Abstract

Thermal analysis is an important topic in many fields, such as building, machinery, and microelectronics. As the types of materials in a system are increasingly diverse, conventional numerical methods or machine learning-based surrogate models face tremendous challenges in computation cost and accuracy. Furthermore, a realistic system usually suffers from random fabrication variations that induce significant errors in model prediction. To overcome these issues, we propose Graph Neural Networks (GNN) as a framework for thermal analysis of composite materials with diverse thermal conductivity and thermal interface variations. Using chiplets in microelectronics as the study case, we first partition the system into sub-blocks based on their material property. Then we develop a physics-constrained GNN as the aggregator to integrate local models of each sub-block into a system, with the edge to represent the thermal interaction. In the presence of interface variations, we introduce continual adaptation of the GNN model, using a minimum number of training samples. Compared with previous solutions, our GNN model is robust for various material and interface conditions, and efficient in the prediction of hot-spot. All codes are publicly available at `https://github.com/thermalanalysis/iclr`

## 1 Introduction

Thermal analysis is of great importance across numerous industrial domains, such as material characterization, environmental testing, electronics design and manufacturing. In microelectronics, an efficient and precise thermal prediction helps ensure the chip meet various specifications at each stage of the design process. The traditional finite element method (FEM), while capable of delivering highly accurate results, is burdened by its substantial computational cost Huang & Usmani (2012). Numerous machine learning (ML) methods, such as physics-informed neural network (PINN) Raissi et al. (2019), have emerged as the lightweight alternatives to FEM. However, these methods are restricted to scenarios involving only uniform or dual-material designs, which are not well-suited for a realistic complex system.

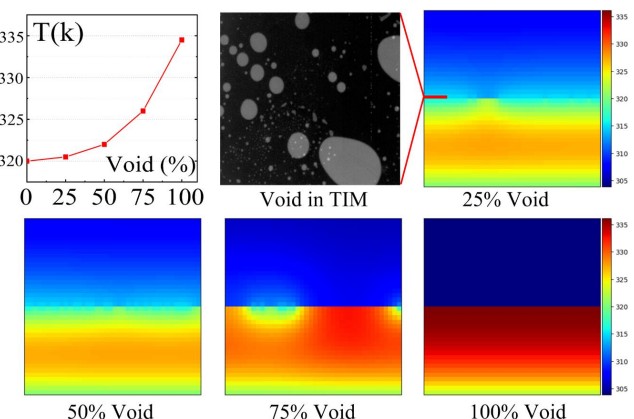

Figure 1: Top-Left: Heat accumulation with more voids in TIM. Top-Middle: Real photo of TIM defects Okereke & Ling (2018). The Rest: Thermal conduction under different ratio of defects in TIM.

Moreover, both FEM and ML-based methods encounter two common challenges. Firstly, whenever a design parameter undergoes a change, even if it is a minor localized change, such as introducing a

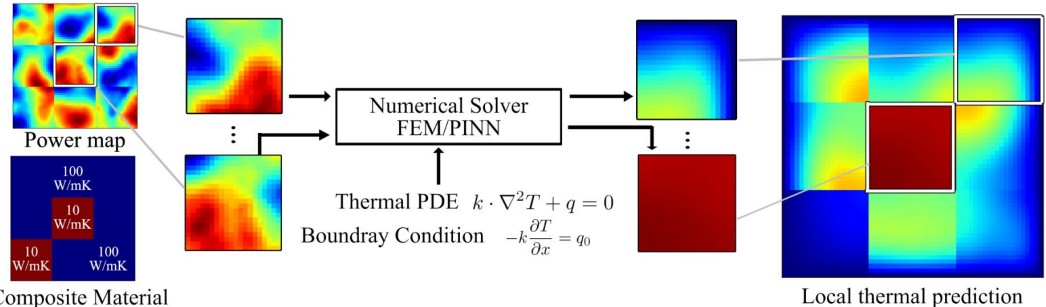

Figure 2: Block Decomposition: Divide the block into small sub-blocks, each region with a uniform material. Evaluate the thermal solution locally using the numerical solver.

new power profile, the previous thermal analysis becomes obsolete, requiring another round of comprehensive full-scale analysis. Secondly, these methods lack the flexibility to address the interface defects introduced by the manufacturing process, For instance, numerous studies (Okereke & Ling (2018); Due & Robinson (2013); Ramos-Alvarado et al. (2013); Otiaba et al. (2014)) have investigated the impact of voids appearing in thermal interface material (TIM), a thin layer between the silicon die and the heat spreader to enhance the heat conduction. As shown in Fig. 1, the presence of these voids significantly reduces the efficiency of thermal transfer between two materials, thereby compromising the reliability of the system. Besides, due to the randomness in the shape, location, and intensity of these voids, each individual system may exhibit unique patterns of defects. Neither FEM nor ML methods can rapidly adapt to such random variation and regenerate an updated thermal output, resulting in a lack of robust analysis tools to predict the actual thermal distribution within flawed yet more practical scenarios.

To overcome the challenges mentioned above, we propose to use the physics-constrained graph neural network (GNN) to conduct thermal analysis on the composite materials, with continual learning of GNN to capture the impact of random variations. Our contributions are as follows:

- Block Decomposition: A domain decomposition/aggregation strategy by partitioning a composite structure into sub-blocks based on their material properties, enabling the numerical solver to efficiently produce thermal solutions at the sub-block (local) level.

- Thermal Aggregation: A physics-constrained GNN to aggregate and connect the local solutions by embedding the physics law of thermal transfer into the message-passing interaction of the graph.

- Continual Graph Learning: A continuous adaptation approach by adding trainable nodes into the graph. These nodes represents random defects with unknown conductivity, such as the interface voids in Fig. 1. With minimal training overhead and without the need to retrain the GNN model, our framework quickly adapts to the change caused by the defects and regenerate accurate thermal prediction.

## 2 BACKGROUND

**Physics-Constrained GNN**   GNN Scarselli et al. (2008) has received an increased interest in characterizing the physical laws that govern the dynamics of particle-based systems. For instance, Sanchez-Gonzalez et al. (2020) propose Graph Network-based Simulators (GNS) as surrogate models for fluid simulation through a message-passing strategy. Pfaff et al. (2020) further enhances the GNS by incorporating both mesh-space and world-space messages into the node and edge embeddings. Besides that, Li et al. (2022); Allen et al. (2023); Bhattoo et al. (2022); Hernández et al. (2022) all prove the GNN's ability to generalize the physical laws across diverse domains, including robotics simulators, molecular dynamics, and rigid body dynamics. Despite the wide application of GNN in physics, thermal analysis has not been extensively explored.

**Thermal Analysis**   To address the high computational cost associated with FEM, several ML-based algorithms have emerged as alternatives. Proposed in Raissi et al. (2019), PINN aims to

Figure 3: Thermal Aggregation: Given the local solution for each sub-block from the block decomposition step, model thermal transfer between sub-blocks through message passing.

discover solutions for a partial differential equation (PDE) by leveraging the benefits of automatic differentiation in machine learning. It has been utilized for solving thermal heat transfer problems in Cai et al. (2020); Hennigh et al. (2021). However, these studies typically focus their analysis on a uniform material. Tod et al. (2021) proposes a dual PINN model for a tool-composite system, incorporating additional loss terms to constrain the heat flux across the interface. However, the scalability to handle more complex systems with multiple materials becomes challenging due to the exploding number of loss terms.

Furthermore, the thermal solutions derived from PINN are constrained by the specific power maps they were trained on, which limits their utility when generalized to unseen cases. To overcome this issue. Liu et al. (2023) applies DeepONets Lu et al. (2021) to learn the function operators such that it can take arbitrary PDE configurations as the function inputs. He & Pathak (2020); Ranade et al. (2022) utilize AutoEncoder (AE) to handle unseen power maps. Nevertheless, Liu et al. (2023); Ranade et al. (2022) suffers from scalability issues due to its model complexity. He & Pathak (2020) lacks the necessary physical constraints governed by PDE.

Yang et al. (2023); Sanchis-Alepuz & Stipsitz (2022) use similar approaches by applying physics-constrained GNN to transient thermal analysis. To improve the interpretability of the GNN, Yang et al. (2023) proposes to use RC models rather than a trainable function to regulate thermal flow. Sanchis-Alepuz & Stipsitz (2022) introduces an additional layer of physical calculations involving material density and heat capacity. However, both approaches consider each node as representing an individual thermal point rather than a block, which makes their method less effective in the analysis of a complex system.

## 3 METHODOLOGY

As shown in Algorithm 1, our framework comprises three main components: (1) Block Decomposition, (2) Thermal Aggregation using GNN, and (3) Continual Graph Learning. We will begin by providing an overview of each step in section 3.1 and subsequently delve into the implementation details in section 3.2.

### 3.1 FRAMEWORK

**Block Decomposition.** As shown in Fig. 2, we take a complex 2D microelectronic structure, such as chiplets, as our study case and partition it into $N$ sub-blocks. The criteria is that each sub-block is either sufficiently small in size or contains a uniform material. This enable us to conduct swift thermal analysis using a numerical solver. At this step, each block is not aware of its adjacent blocks, thus any new boundary created by this partition is thermally isolated. In other words, we place a barrier in between any two sub-blocks to prevent the thermal exchange of heat. Convection boundary conditions are exclusively assigned to the sub-blocks adjacent to the actual boundary.

This decomposition offers us two advantages. Firstly, it provides the flexibility to perform partial thermal analysis while keeping the previously computed local solutions intact. Instead of conducting thermal analysis at the full chip scale, we can employ this "divide and conquer" strategy to reuse existing solutions. Secondly, through material-based decomposition, we can maximize the utility of efficient PINN methods for analyzing uniform materials. For instance, we can train two Deep-

---

**Algorithm 1** Thermal Decomposition/Aggregation and Graph Continual Learning

---

1: **function** $ChipDecomposition\,(designs, defects)$
2:     **for each** $D_i$ in $designs$ **do**
3:         Decompose $D_i$ into $N$ subblocks $\mathcal{D}_i^{local} = \{(p_j, k_j), j \in [0, N)\}$
4:         **for each** $d_j = (p_j, k_j)$ in $\mathcal{D}_i^{local}$ **do**
5:             $t_j \leftarrow Numerical\_Solver(d_j)$
6:         **end for**
7:         Convert $T_i^{local} = \{t_j, j \in [0, N)\}$ to $G_i = (\mathcal{V}, \mathcal{E})$
8:         $T_i^{global} \leftarrow Numerical\_Solver(D_i)$
9:         $T_i^{novel} \leftarrow Numerical\_Solver(D_i, defects)$
10:     **end for**
11:     **return** $\mathcal{G} = \{G_i\}, \mathcal{T}_{global} = \{T_i^{global}\}, \mathcal{T}_{novel} = \{T_i^{novel}\}, i \in [0, \#designs)$
12: **end function**

13: ---

14: **function** $Continual\_Learning(\mathcal{G}_{train}, \mathcal{T}_{novel}, \mathcal{M})$
15:     $\hat{\mathcal{T}} \leftarrow \mathcal{M}(\mathcal{G}_{train})$
16:     $novel\_loc \leftarrow$ where $Abs(\hat{\mathcal{T}} - \mathcal{T}_{novel}) > threshold$
17:     $\mathcal{G}'_{train} \leftarrow$ Insert defects nodes $\mathcal{V}_{new}$ into each $G_{train} \in \mathcal{G}_{train}$ at location $novel\_loc$
18:     Train $\mathcal{V}_{new}$ by $\partial(MSE(\mathcal{M}(\mathcal{G}'_{train})), \mathcal{T}_{novel})/\partial\mathcal{V}_{new}$
19:     **return** $\mathcal{V}_{new}, novel\_loc$
20: **end function**

21: ---

22: **function** $Main\,(designs, defects)$
23:     $\mathcal{G}_x, \mathcal{T}_y, \mathcal{T}_y^{novel} \leftarrow ChipDecomposition(designs, defects)$
24:     $\mathcal{G}_x^{train}, \mathcal{T}_y^{train}, \mathcal{T}_{y\_novel}^{train} \leftarrow \mathcal{G}_x[:N_{train}], \mathcal{T}_y[:N_{train}], \mathcal{T}_y^{novel}[:N_{train}]$
25:     $\mathcal{G}_x^{test}, \mathcal{T}_y^{test}, \mathcal{T}_{y\_novel}^{test} \leftarrow \mathcal{G}_x[N_{train}:], \mathcal{T}_y[N_{train}:], \mathcal{T}_y^{novel}[N_{train}:]$
26:     Train GNN $\mathcal{M}$ using $\mathcal{G}_x^{train}, \mathcal{T}_y^{train}$
27:     Evaluate $\mathcal{M}$ using $\mathcal{G}_x^{test}, \mathcal{T}_y^{test}$
28:     $\mathcal{V}_{new}, loc\_v \leftarrow Continual\_Learning(\mathcal{G}_x^{train}, \mathcal{T}_{y\_novel}^{train}, \mathcal{M})$
29:     $\mathcal{G}_{x\_novel}^{test} \leftarrow$ Insert defects nodes $\mathcal{V}_{new}$ into each $G_x^{test} \in \mathcal{G}_x^{test}$ at location $loc\_v$
30:     Evaluate $\mathcal{M}$ using $\mathcal{G}_{x\_novel}^{test}, \mathcal{T}_{y\_novel}^{test}$
31: **end function**

---

OHeat Liu et al. (2023) models for a dual-material design. All sub-blocks can be sent to these two models based on their materials for efficient thermal prediction.

**Thermal Aggregation.** Given the power map and the materials of each sub-block, we collect the local thermal predictions from the numerical solver, denoted as $\mathcal{T} = \{T_1, T_2, \ldots, T_N\}$, concatenated them with the corresponding power map input and the material conductivity to form the sub-block features $\mathcal{V} = \{T_i \| P_i \| k_i, i \in [0, N)\}$. Based on the physical location of each sub-block, the subdivided chiplet can be regarded as an undirected graph $G = \{\mathcal{V}, \mathcal{E}\}$, with each node $v_i \in \mathcal{V}$ representing a sub-block and its neighbors $\mathcal{N}_i$ consisting of the adjacent sub-blocks that connected by edge $e_{ij} \in \mathcal{E}$. Besides, each edge $e_{ij}$ is assigned a binary encoded ID to characterize the four directions in 2D plane based on the relative location of $v_i$ and $v_j$. The entire graph is then sent to the GNN for thermal aggregation $G \rightarrow \hat{G}$. The output $\hat{G} = \{\hat{\mathcal{V}}, \mathcal{E}\}$ will be in the same structure of $G$ with each nodes $\hat{v}_i \in \hat{\mathcal{V}}$ contains the prediction thermal $\hat{T}_i$. Intuitively, we eliminate the barriers that obstruct thermal exchange and employ GNN to conduct message-passing strategy for thermal exchange. The primary task is to train a GNN using dataset $\{\mathcal{G}_{local}^{Train}, \mathcal{T}_{global}^{Train}\}$ collected from the numerical solver, and deploy it to the unseen testing case to check its aggregation accuracy. Ideally, the testing prediction obtained through this domain aggregation strategy should match the golden results achieved when considering the entire chiplet as one large, unified block. Further detail regarding the structure of GNN will be discussed in section 3.2.1.

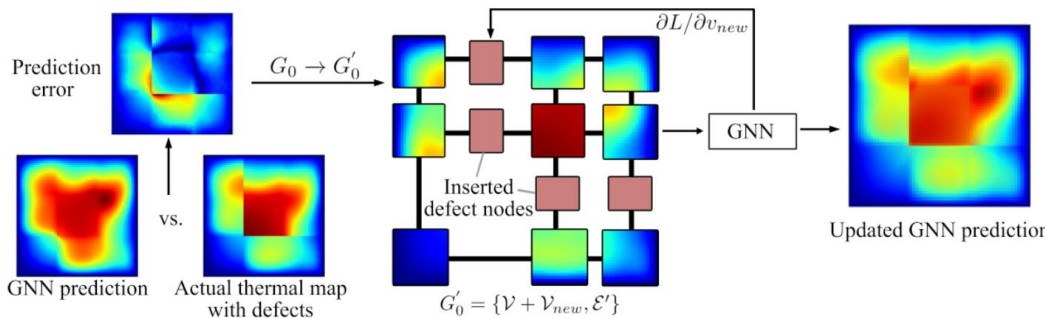

Figure 4: Continual Graph Learning: Insert defect nodes at the locations where GNN prediction significantly differs from the actual thermal measurement. The defect nodes are trained with low volume of training data. Other GNN nodes and edges trained from the previous step is preserved.

**Continual Graph Learning.** Following the establishment of a GNN in the previous step, our next objective is to study how to effectively adapt this GNN to random variations. This includes scenarios where the actual chip may have defects or impurities resulting from fabrication variations, thus impeding the thermal exchange at the specific regions. We assume our block decomposition has a fine enough granularity so that these defects may exist only at the boundaries of each individual block. Prior research (Okereke & Ling (2018); Due & Robinson (2013); Ramos-Alvarado et al. (2013); Otiaba et al. (2014)) also indicated that these manufacturing defects are more likely to occur in the regions adjacent to the interfaces between two materials. Therefore the thermal prediction of the local sub-block will remain the same, and the input graph $G$ to the GNN is not affected as well. But the prediction output $\hat{G}$ from GNN will differ from the golden thermal solution, as the latter reflect the presence of hotspots caused by the defects. The goal is to perform a continual graph learning with minimal training overhead so that the new prediction $\hat{G}$ can also account for the presence of those hotspots. We will address this problem in section 3.2.2

## 3.2 Physics-constrained Graph Network

### 3.2.1 Thermal Analysis

As shown in Fig. 3, our proposed GNN is an adapted version of the previously introduced GNS Sanchez-Gonzalez et al. (2020), comprising an encoder, processor, and decoder. In our case, we eliminate the decoder and integrate the encoder into the processor. We apply multi-steps of message-passing through a chain of processors. At each step, the nodes and edges are partially encoded again by using the previous intermediate output and the original edge ID. Since those edge IDs don't closely adhere to any physical rules but as a auxiliary purpose, the re-encoding prevents this artificial information from propagating through the message-passing.

**Processor.** A processor $\hat{G}_{i-1} \rightarrow \hat{G}_i$ is a GNN layer conducting single-step thermal exchange. It consists of the a edge encoder $f_e$, a node encoder $f_v$ and one aggregator $Agg(x_e)$. For each node $v_i$, we first obtain the edge embedding $x_e^{ij} = f_e(v_i, v_j, e_{ij})$, where $v_j$ is the attribute of $v_i$'s neighbour and $e_{ij}$ is the edge ID. This setup is considered as evaluating the thermal contribution from the neighbouring node to the center node. Next we apply the aggregator to gather these contributions and concatenate them with the center node's features to generate the node embedding $\hat{T}_i = f_v(v_i, Agg(x_e^i))$, which is the output of the processor. The encoder functions $f_e$ and $f_v$ are trainable function, implemented by MLP. $Agg(x_e^i)$ function is the element-wise sum of the edge embeddings $x_e^{ij}$ for all edges connecting to the node $i$.

**Physical Interpretation.** Thermal PDE is written as

$$k \cdot \nabla^2 T + q = 0 \tag{1}$$

where $k$ is the conductivity, $q$ is the heat source term. The edge of the block is governed by the heat flux boundary condition

$$-k\frac{\partial T}{\partial x} = q_0 \tag{2}$$

where $q_0$ is the surface heat flux. When $q_0$ equals zero, the surface is entirely isolated, analogous to the boundary wall we introduce between local blocks during chip decomposition. For a single sub-block that contains four neighbors, given power map $q$, conductivity $k$, and four boundary conditions in the shape of (2), a fixed thermal solution can be determined. Under this context, we can consider our edge embedding function $f_e$ as the process of finding those four boundary conditions. And the aggregator $Agg(x_e)$ and node encoder $f_v$ are together as the PDE solver to take the initial thermal $\hat{T}_{i-1}$ and all boundary conditions to find the thermal solution.

### 3.2.2 CONTINUAL GRAPH LEARNING

Given the randomness of the location and magnitude of each variation, it is infeasible to conduct a separate thermal analysis for every individual case. Even with our proposed GNN approach, the process of preparing new training samples to retrain the GNN incurs a significant amount of computational and storage costs, which can offset the advantages of obtaining highly accurate thermal predictions. Considering that our trained GNN has already demonstrated the capability to generalize the physical laws governing thermal transfer, we propose to preserve the GNN while altering the input graph $\mathcal{G}$ to accommodate the novel behaviors of the materials.

As shown in Fig. 4 and Algorithm 1, we propose to insert trainable defects nodes $\mathcal{V}_{new}$ into $\mathcal{G}$ at the location where abnormal behaviors emerge. For example, if a thermal barricade is detected between node $v_i$ and $v_j$, a new defect node $v_k$ will be placed in the middle to represent the transitional materials with unknown conductivity, and the edge $e_{ij}$ will be broken into $e_{ik}$ and $e_{kj}$. These new nodes represent the anomalous thermal exchanges between the original two nodes. Following this pattern, we take a small amount of the original training samples from $\mathcal{G}_{local}^{Train}$, insert the physical nodes $\mathcal{V}_{new}$ at the consistent location of each graph $G \in \mathcal{G}_{local}^{Train}$. Then we send the updated training dataset, $\{\mathcal{G}', \mathcal{T}_{global}^{novel}\}$ to GNN for nodes training by conducting back-propagation with the gradient respect to $\mathcal{V}_{new}$. This training procedure can be viewed as the process of finding the hidden conductivity of $\mathcal{V}_{new}$.

Once the training converges, we insert $\mathcal{V}_{new}$ into any testing graphs before forwarding them to the GNN. Subsequent thermal predictions will then incorporate this learned material, resulting in outputs that accurately reflect the abnormal thermal behavior.

## 4 EXPERIMENT

In this section, we first test the aggregation performance of our proposed framework using unseen test samples encompassing various combinations of materials. While keeping the GNN fixed, we subsequently introduce defects into the previous training and testing dataset and apply our continuous learning strategy to accommodate these variations. Finally, we compare the efficiency of our algorithm with traditional FEM across varying levels of task complexity. All experiments are implemented using PyTorch Paszke et al. (2019) on NVIDIA GeForce RTX A6000 platform. The prediction results are compared with the numerical solution obtained from the FEM solver.

**Experiment Setup.** For the composite material task in experiment 4.1 and 4.2, we test our algorithm in a 3D 1mm×1mm×0.5mm block with the resolution of 100×100×1. We further decompose it into a 5×5 grid in $x, y$ dimension, with each sub-block having the resolution of 20×20. We assume each sub-block consists of uniform material and is randomly assigned a conductivity value chosen from the options of $1W/(mK)$, $10W/(mK)$, $50W/mK$ and $100W/(mK)$. For the uniform material task in experiment 4.1, we conduct tests with various resolutions and different sub-block sizes. The specific values are provided in the caption of Fig. 6. For most sub-blocks in these setups, we assign them with a 2D power map generated by the Gaussian random field and normalize the power values within the range of 0 to $1mW$. The edges and the bottom of the block are given a fixed $T_{amb} = 298K$ boundary condition. All results presented below are on the power map layer.

**Dataset Preparation.** We generate 400 blocks with random power map inputs and conductivities. Out of these blocks, 320 are allocated for training, 60 for validation, and 20 for testing purposes. For each block, we use the FEM solver to generate both local and global solutions. The local results are transformed into the graph input for the GNN. The global solutions obtained from the numerical solver serve as the reference for calculating losses and evaluating accuracy.

For the purpose of continual learning, we utilize the same power map and conductivity configuration as in the previous experiment. We then perform a new thermal analysis at the global level using the FEM solver. This time, we manually insert heat isolation walls into the blocks to simulate interface defects that obstruct thermal flux. These new evaluations are viewed as a variation of the previous golden one. Out of the 400 blocks mentioned earlier, we only employ a subset of them for the purpose of continual learning. The performance of using different numbers of training samples is analyzed in section 4.2.

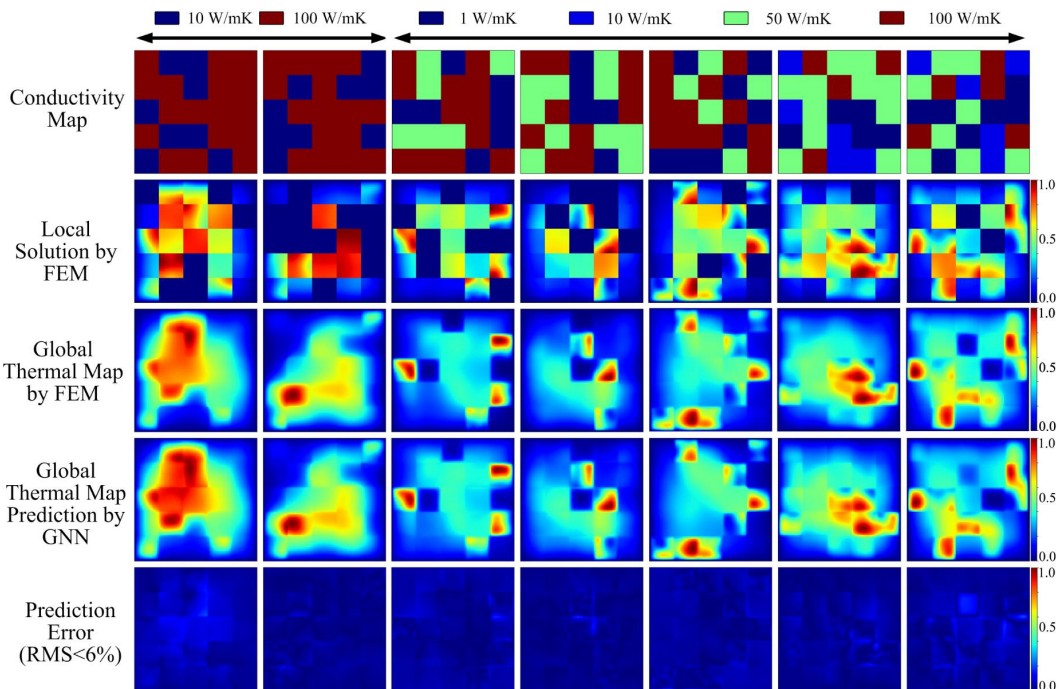

Figure 5: Comparison of our prediction results with the reference solutions across different combinations of materials and power map inputs.

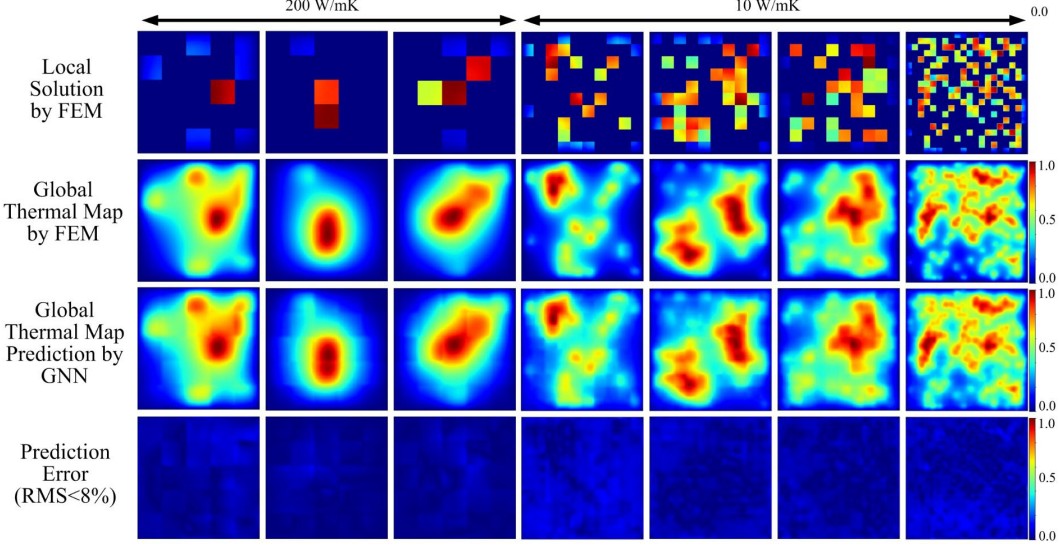

Figure 6: Comparison of our prediction results with the reference solutions on uniform material with different levels of resolution. Sub-block resolution setup from left to right: $20 \times 20$ for all 200W/mK cases, $10 \times 10$ for columns 4-6, and $5 \times 5$ for column 7. All block resolutions are set to $100 \times 100$.

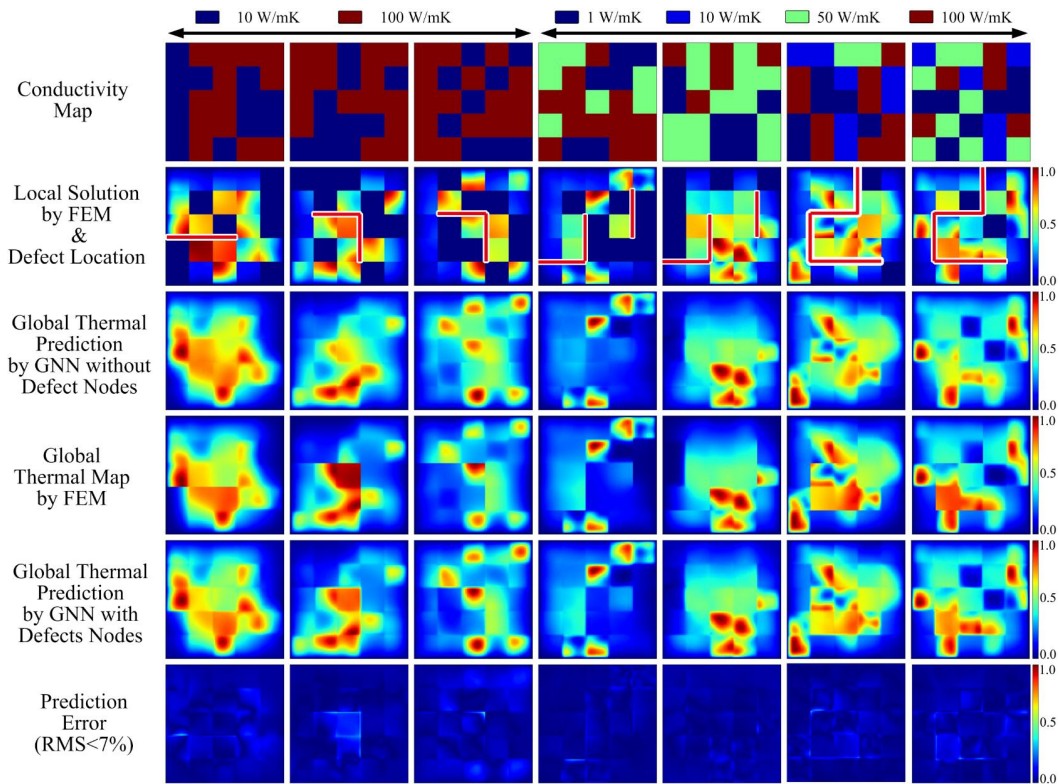

Figure 7: Performance of continual graph learning after inserting defect (highlighted in red color) into the block.

**Graph Neural Network Structure and Training.** Our GNN consists of three processors. The node and edge encoders in each processor share the same three-layer MLP structure, with 512 neurons in the first two layers and 256 neurons in the output layer. All activation functions are configured as Tanh. We employ the Adam optimizer with a learning rate of 0.0001 and weight decay of 0.00005 for both the initial training on the GNN and the continual training on defect nodes. Both training phases encompass 2000 epochs to ensure convergence, with the mean square error serving as the loss function.

## 4.1 PERFORMANCE ANALYSIS OF GRAPH AGGREGATION

As shown in Fig. 5 and Fig. 6, our proposed algorithm demonstrates stable thermal aggregation across different tasks involving blocks with uniform and composite materials. Particularly in scenarios involving materials with high conductivity, such as $50W/mK$ and $100W/mK$, our method effectively simulates the thermal flux across multiple sub-blocks. For example, as shown in the second column of Fig. 5, our method can predict the thermal flow across the left and center of the block, considering the heat source located at the bottom and right side. Furthermore, in the case of materials with low conductivity, our model can accurately represent the block's inability to dissipate thermal energy, resulting in the formation of a hot spot. Our GNN also adheres to the physics at the boundary of the block, as it prevents further thermal transfer when approaching the edge. This demonstrates that the GNN effectively incorporates the edge of the graph into the thermal flux. As the sub-blocks near the boundary lack one or two thermal flux contributions from neighboring nodes, the edge or corner regions retain their original thermal distribution.

## 4.2 CONTINUAL GRAPH LEARNING

As indicated in red lines in the second row of Fig. 7, we evaluate the performance of continual learning by manually introducing thermal blockage to simulate defects with varying levels of difficulty.

We employ straight-line blockages to verify whether our GNN, with the inserted defects nodes in the input graph, can predict the obstruction of thermal flow. Subsequently, we introduce some turns in the blockages to observe if the corner regions can accumulate heat and generate hot spots. Fig. 7 illustrates that our algorithm can correctly handle these tasks. Furthermore, based on the experimental results, our method is not limited to defects occurring at the interface of two materials. It can also accurately handle defects that occur within a single material.

Table 1: Mean absolute percentage errors with different training samples used for continual learning

| MAPE | Training Samples | | | | |
|---|---|---|---|---|---|
| Experiments | 100 | 50 | 20 | 10 | 5 |
| Case1: (10, 100) W/mK | 0.027 | 0.028 | 0.031 | 0.072 | 0.188 |
| Case2: (1, 10, 50, 100) W/mK | 0.038 | 0.041 | 0.057 | 0.113 | 0.232 |

All results shown in Fig. 7 use 20 training samples to train the defect nodes. Table 1 displays the Mean Absolute Percentage Errors (MAPEs) for various training sample sizes. It turns out that training the defect nodes incurs very little training cost while achieving good performance. Compared to the timing and memory cost of retraining the GNN to adapt to every new variation, the use of defect nodes is much more flexible and memory-efficient.

## 4.3 SPEED UP IN THERMAL PREDICTION

Our framework utilizes the classic "divide and conquer" strategy by first calculating the local solution using a numerical solver and then combining the result using GNN, which improves the time complexity from $O(n^2)$ to $O(nlogn)$. Table 2 in rows one and two show the time cost comparison for $N \times N$ blocks between using FEM and our framework. All sub-blocks are set to the resolution of $50 \times 50$. The average time consumption grows exponentially in FEM but remains stable in our approach. Especially in the case of $2500 \times 2500$, our method achieves a $50\times$ speedup.

Additionally, in response to design changes, our proposed algorithm offers users flexibility by requiring only the re-evaluation of a single block from the numerical solver. The updated result from this sub-block is then incorporated with the previous analyses within the GNN framework to generate the new solution. Our approach significantly enhances workflow efficiency and reduces turnaround times as shown in row three of Table 2.

Table 2: Computation time of thermal prediction by (1) Using FEM only for the entire block, (2) Using FEM for all sub-blocks and GNN for merging, and (3) Using FEM for a single sub-block and GNN for merging the new local updated solution.

| Computation Time (s) | Block Resolution $N \times N$ | | | | Speed up |
|---|---|---|---|---|---|
| Experiments | $50\times50$ | $250\times250$ | $500\times500$ | $2500\times2500$ | $2500\times2500$ |
| FEM | 0.004 | 0.753 | 3.971 | 579.09 | 1 |
| All sub-block FEM analysis + GNN | 0.547 | 0.653 | 0.975 | 9.057 | **50×** |
| Single sub-block FEM update + GNN | 0.547 | 0.558 | 0.579 | 1.091 | **500×** |

## 5 CONCLUSION

In this paper, we present a framework for conducting thermal analysis on composite materials through a block decomposition and graph aggregation strategy. Our approach leverages the message-passing capabilities of a physics-constrained GNN to predict thermal interactions between sub-blocks. Tested across various tasks with different material combinations and block resolutions, our method consistently achieves superior prediction accuracy. Additionally, we introduce a continual graph learning method by incorporating trainable defect nodes to represent voids that may appear at material interfaces. With minimal training overhead, our framework rapidly adapts to variations and produces highly accurate thermal predictions.

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
