# OpenReview forum: "Continual Graph Learning for Thermal Analysis of Composite Materials under Interface Variations"
_ICLR.cc/2024/Conference — Submitted to ICLR 2024_

### Official Review · Reviewer_EZT6 · 2023-10-26

**Soundness:** 3 good
**Presentation:** 2 fair
**Contribution:** 1 poor
**Rating:** 5
**Confidence:** 4

**Summary:**

## Summary

The paper proposes a Graph Neural Network (GNN)-based method for predicting the global temperature distribution on chips, given the local block-wise temperature distributions. The authors introduce a novel "graph continuation training" technique involving defect node insertion to model thermal barriers due to manufacturing defects. While the approach is innovative, it suffers from several significant shortcomings including unclear terminology, lack of field specification, and questionable assumptions, among others. I recommend a borderline reject for this paper.

## Detailed Comments

### 1. Lack of Definition and Terminology

The paper fails to clarify critical terminology such as "power map," "material," and "golden thermal.", etc. The absence of these definitions can make the paper hard to follow for readers unfamiliar with the domain of chip manufacturing.

### 2. Target Application Area

The paper implicitly targets thermal problems in chips but fails to explicitly state this. Specifying the domain is crucial as thermal analysis varies greatly between different fields.

### 3. "Physics-Constrained" Misrepresentation

The term "physics-constrained" is used misleadingly. Typically, this term implies that the model architecture or loss terms satisfy physical laws or Partial Differential Equations (PDEs). The paper only provides an analogy between message passing in GNNs and interface heat transfer, which hardly qualifies as "physics-constrained."

### 4. Overambitious Claim in Table 2's row 3

The paper claims that the proposed method shows great acceleration if only one block is updated (in terms of material or heat source). This claim lacks grounding since, in real-world scenarios, it's unlikely that only one block of a chip would be updated independently.

### 5. Limitations of Defect Node Insertion

The defect node insertion strategy, while reasonable for training, is not practical for a new chip where defect locations may not be known beforehand. This limits the method's applicability to the same material type with varying heat sources.

### 6. Benchmarking Issues

The paper compares its performance against solvers that are seemingly not state-of-the-art. For example, multi-grid-based solvers are known to perform well for problems like these and can beat the O(N^2) time complexity mentioned. Additionally, the time reported for Finite Element Method (FEM) solvers seems too slow, possibly indicating the use of a sub-optimal solver.

The authors should reveal more details on the FEM solver implementation.

## Conclusion

The paper introduces a novel approach for predicting the global temperature distribution on chips using GNNs and a unique training strategy. However, it suffers from various issues ranging from unclear definitions and assumptions to flawed benchmarking. These issues severely limit the paper's impact and reliability. Therefore, I recommend a borderline reject for this submission.

**Strengths:**

See above in the summary

**Weaknesses:**

See above in the summary

**Questions:**

See above in the summary

---

> ### Author Response · Authors · 2023-11-23
> **Response to Reviewer EZT6**
>
> We sincerely thank you for your time and efforts. Please find the response to each comment.
>
>
> > **The paper fails to clarify critical terminology such as "power map," "material," and "golden thermal.", etc. The absence of these definitions can make the paper hard to follow for readers unfamiliar with the domain of chip manufacturing.**
>
> Thanks for the suggestion. We agree that these terminologies need to be more clearly addressed. "Power Map" gives the info of power consumption for each blocks. A higher power consumption will contribute to a hotspot. "Material" is related to the conductivity value. Some materials, such as air and substrate, have low conductivity thus more likely to generate hotspot.  "Golden Thermal" refers to the target Y in machine learning which is generated by FEM solver. It is the golden truth given the power map and conductivity value.
>
>
> > **The paper implicitly targets thermal problems in chips but fails to explicitly state this. Specifying the domain is crucial as thermal analysis varies greatly between different fields.**
>
> Our method is not limited to chip applications; it can be applied to any system requiring static thermal analysis. For instance, it can predict the heat dissipation of a building based on room temperatures and their respective locations.  Then we can convert these rooms into a graph and send it to GNN for thermal prediction.
>
>
> > **The term "physics-constrained" is used misleadingly. Typically, this term implies that the model architecture or loss terms satisfy physical laws or Partial Differential Equations (PDEs). The paper only provides an analogy between message passing in GNNs and interface heat transfer, which hardly qualifies as "physics-constrained."**
>
> We agree that the term "physics-constrained" may be too strong and we will refrain from using it in the next version. However, we did not treat the GNN as a blackbox to simply fit the data but carefully design the node/edge encoder and aggregator to mimic the actual physical system.  As discussed in section 3.2.1, this steps are similar to the precedure of solving PDEs, where we collected the necessary boundary and initial conductions to get the final solutions.
>
> > **The paper claims that the proposed method shows great acceleration if only one block is updated (in terms of material or heat source). This claim lacks grounding since, in real-world scenarios, it's unlikely that only one block of a chip would be updated independently.**
>
> There are some real-world scenarios involving local changes. For example, it's common to have multiple power modes for individual blocks on a chip. Some blocks may be powered on or off based on the current task, which leads to localized temperature variations.
>
>
> > **The defect node insertion strategy, while reasonable for training, is not practical for a new chip where defect locations may not be known beforehand. This limits the method's applicability to the same material type with varying heat sources.**
>
> We don't need to know the locations of defects beforehand. Suppose we have a chip and a trained GNN. prior to the chip's manufacturing, we can conduct theoretical thermal analyses using GNN. This prediction is based on the ideal scenario where no unforeseen defects manifest at any location. However upon the actual manufacturing of the chip, the real thermal measurements may deviate from the earlier predictions due to the presence of unforeseen defects. We need to compare the previous prediction with the actural meansurement to determine where the abnormal locations are and insert the defects node accordingly.
>
>
> > **The paper compares its performance against solvers that are seemingly not state-of-the-art. For example, multi-grid-based solvers are known to perform well for problems like these and can beat the O(N^2) time complexity mentioned. Additionally, the time reported for Finite Element Method (FEM) solvers seems too slow, possibly indicating the use of a sub-optimal solver.**
>
> To the best of our knowledge, the best known complexity for FEM method is O(n^1.8) which is still worse than divide and conquer.  Despite our method requiring FEM calculations for each local block, the time overhead increases linearly as compared to using FEM on full scale analysis which grows exponentially.

---

### Official Review · Reviewer_sWQE · 2023-11-05

**Soundness:** 3 good
**Presentation:** 2 fair
**Contribution:** 2 fair
**Rating:** 3
**Confidence:** 3

**Summary:**

This paper examines thermal analysis for composite materials with non-uniform thermal properties, with graph neural networks and continual learning used to speed up numerical solution obtained from Finite Element Methods (FEM). Citing problems in existing literature with uniform assumptions, the paper proposes a strategy to decompose the material into a number of smaller sub-domains where the heat equation is solved in isolation, followed by aggregation to arrive at the global solution. The aggregation involves message passing and when done through Graph Neural Networks, provides a speedup over a more traditional numerical approach involving FEM, while retaining accuracy.

**Strengths:**

+ Well motivated approach that lends itself to message passing with GNNs
+ Results obtained show that the method is sound.

**Weaknesses:**

- Problem not described very clearly, and related work not comprehensive enough. Graph Neural Networks is not covered thoroughly, in a way that walks the reader through the main developments in the field. I would have hoped that some of the seminal works (e.g. Kipf and Welling [1]) would have warranted a mention.
- Process not described in adequate detail. It is mentioned that the boundary conditions are 'frozen' while solving the sub-domain problem, and then corrected during the aggregation step. This should be explained through equations and a presentation of how the correction takes place, both for the FEM case, and for GNNs.

**Questions:**

Datasets - Did the authors try this on any public datasets that would allow for experimentation? What about comparison with baseline methods?

Aggregation - I would think that a toy problem would illustrate how the aggregation works more clearly.

---

> ### Author Response · Authors · 2023-11-23
> **Response to Reviewer sWQE**
>
> We sincerely thank you for your time and efforts. Please find the response to each comment.
>
>
> > **Problem not described very clearly, and related work not comprehensive enough. Graph Neural Networks is not covered thoroughly, in a way that walks the reader through the main developments in the field. I would have hoped that some of the seminal works (e.g. Kipf and Welling [1]) would have warranted a mention.**
>
>
> Thanks for the suggestion. We will add more references in the background section about the GNNs.
>
> > **Process not described in adequate detail. It is mentioned that the boundary conditions are 'frozen' while solving the sub-domain problem, and then corrected during the aggregation step. This should be explained through equations and a presentation of how the correction takes place, both for the FEM case, and for GNNs.**
>
>
> FEM is only used for local thermal analysis so there is no correction phase. For GNN,
> the edge/node encoder and aggregator we choose help explain how the correction is done. To predict the thermal on each node, the edge encoder and aggregator collects all the neighbouring thermal contributions, and the node encoder utilizes the collected data in conjunction with the node's local thermal information to generate the corrected thermal prediction. As discussed in section 3.2.1, this steps are similar to the precedure of solving PDEs, where we collected the necessary boundary and initial conductions to get the final solutions.
>
> > **Datasets - Did the authors try this on any public datasets that would allow for experimentation? What about comparison with baseline methods?**
>
> We did not test our algorithm on a real chip yet since it requires the thermal camera to capture the temperature. But we created a more complex experiment involving a dummy chiplet that incorporates multiple functional blocks, including AES, SRAM, and EMIB. The conductivity and power of these blocks are extracted from the real physical design data. The chiplet size is 250(L)×230(W)×2.5(H)μm. It is subdivided into 15x13x5 subblocks.  Same as the previous experiments, we randomly generating training samples, incorporating arbitrary power maps and conductivities, to train GNN and then test its performance on this chiplet. The MAPE is ~0.05. We then manually add interface defects at the conjonction of AES and substrate, and apply our continual learning strategy. The updated prediction successfully shows the hotspot at the location of defects with ~0.07 MAPE.
>
>
>
> > **What about comparison with baseline methods?**
>
> We use FEM as the baseline since it provides the most accurate evalution. And to the best of our knowledge, our method is the first ML algorithm tackling the interface varitions problem, therefore there is lack of other methods for comparison.
>
> > **Aggregation - I would think that a toy problem would illustrate how the aggregation works more clearly.**
>
> Please check the test chip experiment mentioned above.

---

### Official Review · Reviewer_bfnq · 2023-11-05

**Soundness:** 2 fair
**Presentation:** 2 fair
**Contribution:** 2 fair
**Rating:** 3
**Confidence:** 3

**Summary:**

The paper proposes to use graph neural networks (GNNs) to solve a thermal PDE for material analysis of chip manufacturing. Specifically, the authors propose to split the image of thermal information into a regular graph, of which each node represents a small patch of the image which can be independently solved with a PDE solver. The final solution of the whole chip can be obtained by passing the information to the GNN for inference. To accommodate noise in the manufacturing process, the authors propose to add artificial defect nodes into the graph structure. The authors present some quantitive and qualitative results to show that their methods are helpful in the application.

**Strengths:**

- The paper tackles an interesting problem of efficiently simulating the thermal properties of wafer, by leveraging graph neural nets (GNNs) for solving heat equation.
- The authors provide some qualitative and quantitive experiment results and showing that GNN is helpful in the downstream task.

**Weaknesses:**

- While the downstream task is definitely interesting, the proposed method is more or less a direct application of existing GNN-based PDE methods to the new problem without much surprise.
- The dataset size is extremely small (320 for training and 20 for testing). I am not sure if such a small dataset size makes sense, especially given that the datasets are generated by simulation (with the simple heat equation) instead of from real-world experiment data.
- I am not sure the finetuning stage with random defect nodes should be called "continual learning". Continual learning, to the best of my knowledge, means the setting where the data can only be accessed from an online data stream and therefore can only be sampled in an non-iid way. I do not see any reason to brand the process of "finetuning with additional noisy data" as continual learning.
- There are many baselines ML methods for efficiently solving PDEs - for instance, Fourier neural operator (FNO). Given that the heat equation is an simple one (of which the Green's function can be found as a simple heat kernel) compared to many other common benchmarks (Burger's eqn / Navier-Strokes eqn, as in FNO and many other papers in the field of PINN), a lack of comparison with some common baselines is detrimental to the quality of the paper.
- The authors do not explain how these "defects" in the dataset are generated. Are they pure Gaussian noises, or something simulated through physical laws?
- The paper states that the "detect nodes" in the graph structure are introduced after checking where the prediction highly differs from the ground truth data. Do the authors imply that the location of the defects are highly predictable? If they are, why? The authors do not explain the reason behind their design.
- The presentation of this paper is not satisfactory: the figures presented in the paper are blurry; background knowledge like what each symbol in the heat equation means is not well explained; the algorithm is presented in a messy way, etc.

**Questions:**

Given these observations, I feel that the paper is not quite ready for submission. I would suggest the authors revise the writing, clearly explain their experiment settings and conduct a comprehensive set of ablation studies to validate their idea.

---

> ### Author Response · Authors · 2023-11-23
> **Response to Reviewer bfnq**
>
> We sincerely thank you for your time and efforts. Please find the response to each comment.
>
> > **The dataset size is extremely small (320 for training and 20 for testing). I am not sure if such a small dataset size makes sense, especially given that the datasets are generated by simulation (with the simple heat equation) instead of from real-world experiment data.**
>
> Despite the small amount of training samples, each sample is a small graph which consists of 25 nodes. The GNN is optimized on a per-node basis so there are 8000 nodes in total for GNN training. We are sorry for the confusion. We will add more detail in the next version.
>
> > **I am not sure the finetuning stage with random defect nodes should be called "continual learning". Continual learning, to the best of my knowledge, means the setting where the data can only be accessed from an online data stream and therefore can only be sampled in an non-iid way. I do not see any reason to brand the process of "finetuning with additional noisy data" as continual learning.**
>
> We fully agree.  Since our algorithm doesn't necessitate retraining the GNN using noisy data in a continual learning pattern, we believe that the term "adaptive learning" would be a more fitting description.
>
> > **There are many baselines ML methods for efficiently solving PDEs - for instance, Fourier neural operator (FNO). Given that the heat equation is an simple one (of which the Green's function can be found as a simple heat kernel) compared to many other common benchmarks (Burger's eqn / Navier-Strokes eqn, as in FNO and many other papers in the field of PINN), a lack of comparison with some common baselines is detrimental to the quality of the paper.**
>
> Our algorithm primarily addresses challenges related to thermal analysis, including the substantial computational costs associated with FEM and the limitations in handling interface variations. Subsequent research could focus on applying GNNs to address other PDEs.
>
> > **The authors do not explain how these "defects" in the dataset are generated. Are they pure Gaussian noises, or something simulated through physical laws?.**
>
> We use FEM method to generate the golden data with "defects" inserted. Specifically, we manually establish a conductance of 0 between certain adjacent nodes, preventing heat exchange between them. FEM can accommodate this zero conductance condition and generate the corresponding thermal map. We are sorry for the confusion and will add more detail in the next version.
>
> > **The paper states that the "detect nodes" in the graph structure are introduced after checking where the prediction highly differs from the ground truth data. Do the authors imply that the location of the defects are highly predictable? If they are, why? The authors do not explain the reason behind their design.**
>
>
> These locations are not predictable. They are detected by comparing the difference between GNN prediction and the actual measurement.
>
> Suppose we have a chip and a trained GNN. prior to the chip's manufacturing, we can conduct theoretical thermal analyses using GNN. This GNN prediction is based on the ideal scenario where no unforeseen defects manifest at any location. However upon the actual manufacturing of the chip, the real thermal measurements may deviate from the earlier predictions due to the presence of unforeseen defects.
> We need to compare the previous prediction with the actural meansurement to determine where the abnormal locations are and insert the defects node accordingly.
> This way, for the same chip, even when a different working load is applied in the future, the trained defect nodes can still be utilized to manage the adjusted predictions.
>
>
> > **The presentation of this paper is not satisfactory: the figures presented in the paper are blurry; background knowledge like what each symbol in the heat equation means is not well explained; the algorithm is presented in a messy way, etc.**
>
> Thanks for pointing out. We will further polish the paper.

---

### Meta-Review · Area_Chair_pJB1 · 2023-12-09

**Metareview:**

This paper presents a GNN-based approach for prediction of thermal properties of materials. While this problem is very interesting, the reviewers agree that the paper has significant issues in terms of presentation and in comparison against baseline methods. I therefore must recommend rejection.

**Justification For Why Not Higher Score:**

The reviewers all recommend rejection.

**Justification For Why Not Lower Score:**

n/a

---

### Decision · Program_Chairs · 2024-01-16

Reject